# An Entropy Dynamics Approach to Inferring Fractal-Order Complexity in the Electromagnetics of Solids

**DOI:** 10.3390/e26121103

**Published:** 2024-12-17

**Authors:** Basanta R. Pahari, William Oates

**Affiliations:** 1Hawai‘i CC Department of Mathematics, University of Hawai‘i, Hilo, HI 96720, USA; basanta@hawaii.edu; 2Department of Mechanical Engineering, Florida Center for Advanced Aero Propulsion, Florida A&M University and Florida State University, Tallahassee, FL 32310, USA

**Keywords:** fractals, entropy dynamics, electromagnetics

## Abstract

A fractal-order entropy dynamics model is developed to create a modified form of Maxwell’s time-dependent electromagnetic equations. The approach uses an information-theoretic method by combining Shannon’s entropy with fractional moment constraints in time and space. Optimization of the cost function leads to a time-dependent Bayesian posterior density that is used to homogenize the electromagnetic fields. Self-consistency between maximizing entropy, inference of Bayesian posterior densities, and a fractal-order version of Maxwell’s equations are developed. We first give a set of relationships for fractal derivative definitions and their relationship to divergence, curl, and Laplacian operators. The fractal-order entropy dynamic framework is then introduced to infer the Bayesian posterior and its application to modeling homogenized electromagnetic fields in solids. The results provide a methodology to help understand complexity from limited electromagnetic data using maximum entropy by formulating a fractal form of Maxwell’s electromagnetic equations.

## 1. Introduction

Entropy dynamics has been used previously to formulate continuum and quantum scale balance equations [1,2]. It is a generalized Bayesian framework that can be used for updating priors given new information cast in the form of constraints to high-fidelity computations or experimental data. While the method is general, Boltzmann and Gaussian distributions are typically considered, which can be obtained by maximizing Shannon’s entropy together with a first-order and second-order moment constraint, respectively [3]. Here, we introduce fractional moments in space and time which allows for a continuum of constraints between the first and second order based on the fractional power-law order ranging from one to two. Other constraints associated with singularities and power-laws are also introduced and related to source terms in Maxwell’s electromagnetic equations. Importantly, relationships are developed that connect the non-integer moments with fractal-order operators based on data contained in the constraints. The results provide a more general representation of electromagnetic field behavior in complex media that cannot be easily approximated using integer-order operators and Gaussian probabilities.

In recent years, fractal and fractional-order operators have garnered significant interest from different scientific communities. Various works on fractional calculus [4,5] and fractal diffusion in materials [6,7] have contributed to a better understanding of how fractal and fractional-order operators offer novel insights into complex material behavior [8,9,10,11,12,13,14,15]. One application of the fractional operator is a novel derivation of fractional-order conservation of mass [16]. In this derivation, the authors demonstrate that when a power-law function can approximate mass flux through porous media, the continuity equation becomes exact if the fractional derivative’s order matches the flux’s power-law and the operating point is about zero mass density. This approach contrasts with the traditional method of deriving conservation of mass, which employs a first-order integer Taylor series to represent flux changes in a control volume. The key reason for using fractional-order operators is that integer Taylor expansions require infinite terms to approximate fractional power-law functions. In contrast, the fractional Taylor series requires only the first-order term. Other studies have used Caputo and other fractional derivatives to couple and analyze nonlinear systems, such as the Jaulent–Miodek equations with an energy-dependent Schrödinger potential, through numerical and semi-analytic methods like the Adomian decomposition technique [17,18]. These approaches effectively tackle complex fractional differential equations, offering valuable insights into nonlinearity and intricate system dynamics, similar to the treatment of Maxwell’s equations presented in this work. Fractional and fractal-order operators are also effective at characterizing the viscoelastic behavior of polymers under different strain rate loading [19,20]. The fractal structure of the polymer network can influence the hyperelastic and viscoelastic constitutive behavior of polymers, and this effect can be better understood using fractional operators [21,22]. In certain instances, the fractal and fractional-order derivatives are equivalent despite the former being a local operator, while fractional-order derivatives are nonlocal [20]. The characteristic scales in space or time will also influence whether fractal or fractional-order operators better approximate state changes [4]. The choice of the appropriate local or nonlocal operator can be related to the form of the probability density. Fractal-order operators are well suited for stretched exponential probability densities, while the fractional Caputo operator is well suited to predict power-law probability densities. It has been argued that the Mittag-Leffler function unifies these different scales such that fractional operators can simulate a broad range of behaviors [23]. Here, we focus on fractal operators due to their local properties, which have many computational advantages in simulating materials as a homogenized continuum. Although fractional calculus has been considered for developing the general electromagnetic field equations [24], less work has focused on using information theory and Bayesian inference as a tool to quantify non-integer derivative orders as a method to formulate continuum-scale electromagnetic models of materials.

Mandelbrot [25] defined fractal geometry as complex, self-similar structures that exhibit intricate patterns at every scale. Unlike traditional geometric shapes, fractals often have non-integer (fractional) dimensions that are characterized by recursive patterns, making them useful for modeling many natural phenomena, including complex materials exposed to electromagnetic fields. Given the pervasive nature of fractals in material structures, researchers have investigated how their heterogeneous characteristics affect electromagnetic fields [26,27,28,29]. Balakin et al. [27] analyzed and applied fractal vector calculus to Maxwell’s electromagnetic equations. Starzewski [29] extended this work to include the effect of anisotropic fractals on electromagnetic fields in complex materials. In the latter, product measures are used to accommodate fractal anisotropy and variational methods are employed to obtain the field equations. From a practical perspective, the fractal design of antennas and metamaterials has been considered to enhance multiband properties, compactness, high directivity, and a stronger understanding of scattering effects due to multiscale surface roughness [30,31].

The importance of stochastic analysis relative to deterministic electromagnetic models has been pointed out in the literature. The papers by Naus [32] and Baker-Jarvis and Surek [33] discuss the application of statistical inference and maximum entropy in understanding electromagnetic fields in the presence of uncertainty. Naus focuses on Gaussian processes and the central limit theorem to evaluate typical polarization and magnetization effects on a continuum scale. Baker-Jarvis and Surek’s approach starts with the assumption of Louisville’s equation as the time-dependent governing equation and applying a projection operator [33]. However, their work focuses on the application of integer-order calculus operators. They also use theoretical constraints in the maximum entropy method, as opposed to experimental observational constraints. In our approach, we do not make distinctions between theory or experimental observations. Either may be integrated into the proposed modeling framework. Furthermore, we generalize the constraints to accommodate extreme events (in terms of fractal-based power-laws) and show how these extreme events map to fractal operators upon statistical homogenization.

The four fundamental equations governing electromagnetic fields in Maxwell’s equations are extraordinarily accurate in material systems where the charge density and current density are well understood [34,35,36]. The distribution of charges and their dynamical properties are typically inferred experimentally or computed for specific thermodynamic states and then homogenized using a variety of approximation procedures [32,33,37,38]. The inference process requires constitutive relationships governing the charge density in Gauss’ law and the current density in Ampere’s law for each particular material. In the case of an ideally amorphous solid or idealized crystalline material, Gaussian statistics prevail, and integer Taylor expansions provide excellent approximations of the charge and current density; see Nelson [37] for details. This is because of the quadratic function within the exponential of the Gaussian distribution, which is correlated with linear constitutive models. In complex materials containing heterogeneities such as point defects, line dislocations, domain walls, and/or grain boundaries, Gaussian probability distributions are less accurate due to the thicker tail nature of the defect distributions. This is further complicated when an electromagnetic solid’s representative volume element (RVE) is reduced to a point. It raises concerns about applying Gaussian statistics and integer-order operators in the equations governing the electromagnetic fields. We propose that fractal operators provide better approximations of electromagnetic fields in materials containing certain non-Gaussian distributions of defects within a representative volume element. Whereas material structure does not completely define material properties, there are well-known relationships between structure and properties. These relationships include anisotropy associated with crystal structure and texture in polycrystalline materials [39]. Less work, however, has focused on understanding the structure-property relationships for fractal structure and non-Gaussian distributions in electromagnetic solids.

Falconer showed that integer diffusion in 2D can be achieved by homogenizing specific observations of the scalar field u(x,t) across the spatial domain x at a given time $t=t_{0}+\Delta t$ [40]. More precisely, if we consider $u(X,t_{0})$, written as a function of the undeformed Lagrangian frame X [41], homogenized over a Gaussian density as
(1)$$u\left(x,t\right)=\frac{1}{\sqrt{4D\pi t}}\int_{\Omega_{0}}e^{-\frac{\left(x-X\right)\cdot \left(x-X\right)}{4Dt}}u\left(X,t_{0}\right)\;dX$$
and by taking a first-order derivative with respect to *t* and the second-order derivative with respect to x on u(x,t), it is well known that we obtain the integer diffusion equation
(2)$$\frac{\partial u}{\partial t}=D\nabla^{2}u.$$
where *D* is the diffusion coefficient and $\nabla^{2}=\frac{\partial }{\partial x}\cdot \frac{\partial }{\partial x}$ is the Laplacian operator.

The inference of continuum-scale material properties from Gaussian density homogenization can lead to uncertainties associated with the application of integer-order operators based on the above relations (also see pp. 270–273 [40]). In cases of rare events such as the onset of dislocations, microcracking, or nucleation of a phase transition, non-Gaussian densities of continuum-scale representative volume elements can provide insight into constructing more accurate space–time operators governing complex and nonlinear material behavior.

Given these complexities in solid materials, we explore the application of an information-theoretic model to infer Bayesian posteriors that guide the formulation of continuum-scale versions of Maxwell’s equations using constraints from observations. In principle, these observations may be in the form of experimental measurements or high-fidelity atomistic or quantum simulations. Importantly, we show how fractional moments in space and time lead to thicker tail probabilities. We then show how fractal operators are self-consistent with the homogenization of local fields over these thicker tail probabilities at the continuum scale. In the limiting case of Gaussian posteriors, the classical integer derivative forms of Maxwell’s equations are consistent with the homogenized equations.

The fractal form of Maxwell’s equations studied here is derived using information theory by maximizing Shannon entropy under a set of fractional moment constraints in space and time. Unlike prior work that integrates over the space of all possible material configurations, integrals over both space and time extend the ability to model diffusion to also include wave propagation. Here, the resulting time-dependent Bayesian posteriors obtained from maximum entropy provide a means to quantify how continuum-scale electromagnetic fields vary in space and time using fractal derivative operators. Previously, we have used this approach to characterize fractal-order constitutive behavior in polymers, such as viscoelasticity, excluded volume effects [19,20] and to estimate molecular dynamics diffusion behavior as a continuum [42], but this method has not been considered for electromagnetic field behavior in solids. This requires extensions to fractal Laplacian and curl operators and fractal wave propagation relations.

The remainder of the paper is outlined as follows. In Section 2, we discuss Hausdorff measures, integration on non-integer-dimensional spaces, fractal derivatives, and the properties of this derivative that are essential for us to establish the combined fractal Maxwell’s equations and entropy dynamics framework. This is followed by the fractal form of Maxwell’s equations in Section 3. In Section 4, we describe how entropy dynamics can be used to infer the fractal form of Maxwell’s equations from sparse data. This section also includes homogenization of the electromagnetic fields using the entropy dynamic inferred Bayesian posterior densities. A simple example illustrating when fractal operators are important is given in Section 4.2, where the fractal conservation of charge is compared to the fractal form of Maxwell’s equations. We also include a discussion of units in Section 4.3, which are known to complicate physical interpretations of fractal operators. Discussion and concluding remarks are given in Section 5, where we summarize the key findings and the practical implications of the methodology on materials science, nanotechnology, and electromagnetic field modeling in heterogeneous media.

## 2. Fractal Measures and Operators

### 2.1. Hausdorff Measure and Integration on Non-Integer Dimensions

Due to the complex nature of fractal or fractional spaces, it is important to define a suitable measure for such spaces. The Hausdorff measure extends the concept of Lebesgue-type measure, and it generalizes notions such as length, area, and volume to encompass more irregular sets that may lack a well-defined dimension, such as fractals [43]. With appropriate measures established, one can then develop other mathematical tools for fractional spaces. Stillinger derived integration rules for certain classes of functions on non-integer dimensional spaces using Hausdorff measures while also introducing the generalized Laplacian operator [44]. We build upon this analysis for applications in electromagnetic field problems.

Consider a measure space (X,M,μ,d), consisting of a set *X*, a collection of Borel subsets *M*, and a metric *d* [45]. The measure μ is taken to be the Hausdorff measure ($\mu_{H}$). For a given set *E*, this measure is defined by
(3)$$\mu_{H}\left(E\right)=\lim_{\delta \to 0}\inf\left\{\sum_{i=1}^{\infty }{\left(\mathrm{diam}\left(U_{i}\right)\right)}^{d}:E\subset \overset{\infty }{\underset{i=1}{⋃}}U_{i},\;\mathrm{diam}\left(U_{i}\right)<\delta \right\}$$
where *d* is a non-negative real number and $\mathrm{diam}(U_{i})$ denotes the diameter of the set $U_{i}$ that forms an open cover of *E*.

The Hausdorff dimension, which usually coincides with the fractal dimension, is then given by
(4)$$\dim_{H}\left(E\right)=\inf\;\left\{d:\mu_{H}\left(E\right)=0\right\}=\sup\;\left\{d:\mu_{H}\left(E\right)=\infty \right\}.$$The fractal integral measure $d\;\mu_{H}\left(x\right)$ is related to integer Lebesgue measure dμ(x) in the following way
(5)$$d\;\mu_{H}\left(x\right)=\frac{2\;\pi^{\alpha /2}}{\Gamma \left(\frac{\alpha }{2}\right)}\;x^{\alpha -1}d\;\mu \left(x\right).$$

For integer $\dim_{H}$, Lebesgue and Hausdorff measures are equivalent [40]. Now suppose $X\subset R^{n}$ is closed, unbounded, and regular such that $\dim_{H}$ is the same or unique overall *X* with respect to the measure given in (Equation 3). Then for a symmetric function *f* we have
(6)$$\int f\left(r\right)\;d\mu_{H}=\sigma \left(D\right)\int f\left(r\right)\;r^{D-1}\;dr.$$
where
(7)$$\sigma \left(D\right)=\frac{2\pi^{\frac{D}{2}}}{\Gamma (\frac{D}{2})}.$$

It is also shown by Palmer and Stavrinon [45] that for a spherically symmetric function $f:R^{n}\to R$,
(8)$$\int f\left(r\right)\;d\mu_{1}\left(x_{1}\right)\;d\mu_{2}\left(x_{2}\right)\cdot \cdot \cdot \;d\mu_{n-1}\left(x_{n-1}\right)\;d\mu_{n}\left(x_{n}\right)=\sigma \left(D\right)\int f\left(r\right)\;r^{D-1}\;dr$$
where $r^{2}=\sum_{i}x_{i}^{2}$ and $D=\sum_{i}{\alpha_{i}}^{2}$.

### 2.2. Fractal Derivatives

This section defines fractal derivatives, divergence, curl, and Laplacian operators and proves some of their basic properties that we apply to electromagnetics. The fractal derivative of order β>0 of a function *f* is
(9)$$\frac{d}{dx^{\beta }}f\left(x\right)=\lim_{x_{0}\to x}\frac{f\left(x_{0}\right)-f\left(x\right)}{{x_{0}}^{\beta }-x^{\beta }}.$$

This derivative differs from the traditional integer derivative in the sense that the spatial parameter is scaled to the power of β. Such scaling is useful in modeling phenomena that exhibit a power-law relationship, as seen in materials following a fractal pattern. This is why it is referred to as the “fractal derivative”. The fractal derivative and the traditional integer derivative share the following relationship
(10)$$\frac{d}{dx^{\beta }}f\left(x\right)=\frac{1}{{\beta |x|}^{\beta -1}}\frac{df(x)}{dx}.$$

The relationship above simplifies the implementation of this fractal derivative and demonstrates its nature as a local operator. Because of its local nature, fractal derivatives lead to numerically faster and more stable algorithms than other nonlocal fractional-order operators, such as the Caputo derivative. Figure 1 shows how the fractal derivatives of sin(x) and $e^{-x^{2}}$ change with changing fractal derivative order. One of the fundamental properties of derivatives is their linearity, which also holds true for fractal derivatives.

The fractal gradient of a function of three variables $f(x_{1},x_{2},x_{3})$ is defined as
(11)$$\begin{array}{c} \nabla_{\beta }f=\frac{\partial f}{\partial x_{1}^{\beta }}i+\frac{\partial f}{\partial x_{2}^{\beta }}j+\frac{\partial f}{\partial x_{3}^{\beta }}k=\frac{1}{\beta |x_{1}{|}^{\beta -1}}\frac{\partial f}{\partial x_{1}}i+\frac{1}{\beta |x_{2}{|}^{\beta -1}}\frac{\partial f}{\partial x_{2}}j+\frac{1}{\beta |x_{3}{|}^{\beta -1}}\frac{\partial f}{\partial x_{3}}k. \end{array}$$

Divergence and curl operators play essential roles in fractal calculus. The fractal divergence and curl of a vector function F are
(12)$$\begin{array}{cc} \nabla_{\beta }\cdot F & =\frac{\partial F_{1}}{\partial x_{1}^{\beta }}+\frac{\partial F_{2}}{\partial x_{2}^{\beta }}+\frac{\partial F_{3}}{\partial x_{3}^{\beta }} \end{array}$$
and,
$$\begin{array}{c} \nabla_{\beta }\times F=\left|\begin{array}{ccc} i & j & k\\ \frac{\partial }{\partial x_{1}^{\beta }} & \frac{\partial }{\partial x_{2}^{\beta }} & \frac{\partial }{\partial x_{3}^{\beta }}\\ F_{1} & F_{2} & F_{3} \end{array}\right|=\left(\frac{\partial F_{3}}{\partial x_{2}^{\beta }}-\frac{\partial F_{2}}{\partial x_{3}^{\beta }}\right)i+\left(\frac{\partial F_{1}}{\partial x_{3}^{\beta }}-\frac{\partial F_{3}}{\partial x_{1}^{\beta }}\right)j+\left(\frac{\partial F_{2}}{\partial x_{1}^{\beta }}-\frac{\partial F_{1}}{\partial x_{2}^{\beta }}\right)k. \end{array}$$

Since the fractal derivative is a linear operator, it can be shown that both fractal divergence and curl exhibit linearity, akin to integer calculus. We will use this linear property to derive the fractal Maxwell’s equations in Section 3. Moreover, the fractal curl of the fractal gradient is also zero, i.e., $\nabla_{\beta }\times \nabla_{\beta }f=0$. In addition, the fractal divergence of the fractal curl is also 0. In other words, $\nabla_{\beta }\cdot \left(\nabla_{\beta }\times F\right)=0.$

The last fractal operator we define in this section is the fractal Laplacian. The fractal Laplacian is
(13)$$\nabla_{\beta }^{2}f=\frac{\partial }{\partial x_{1}^{\beta }}\frac{\partial f}{\partial x_{1}^{\beta }}+\frac{\partial }{\partial x_{2}^{\beta }}\frac{\partial f}{\partial x_{2}^{\beta }}+\frac{\partial }{\partial x_{3}^{\beta }}\frac{\partial f}{\partial x_{3}^{\beta }}.$$

Analogous to its integer counterpart, the fractal divergence of a fractal gradient corresponds to the fractal Laplacian. This operator may also be written as $\nabla_{\beta }\cdot \nabla_{\beta }f=\nabla_{\beta }^{2}f$.

The Taylor series expansion is widely studied and applied in engineering to approximate various quantities of interest in fields ranging from thermodynamics control systems and signal processing. The fractal analog to the integer Taylor series is
(14)$$f\left(x\right)=\sum_{k=0}^{\infty }\frac{1}{k!}{\left(\frac{d}{dx^{\beta }}\right)}^{k}f\left(x=x_{0}\right){\left(x^{\beta }-{x_{0}}^{\beta }\right)}^{k}$$The two-term fractal Taylor series of a power-law function is exact [19]; in contrast, it takes the integer Taylor expansion an infinite number of terms to perfectly approximate a fractional power-law function.

It is possible to extend Falconer’s approach in Equation (Equation 1) to model fractal diffusion processes by homogenizing the stretched exponential distribution. The time-dependent fractal diffusion equation can be derived from the following statistical homogenization relationship for N-dimensional stretched Gaussian probability densities
(15)$$u\left(x,t\right)=\frac{A}{\sqrt{t^{N\alpha }}}\int_{\Omega_{0}}e^{-\frac{\left(x^{\beta }-X^{\beta }\right)\cdot \left(x^{\beta }-X^{\beta }\right)}{4Dt^{\alpha }}}u\left(X,t_{0}\right)\;dX$$
where $\Omega_{0}$ is the region of integration, $A=\frac{1}{\sqrt{{\left(4\pi D\right)}^{N}}}$, α is the fractal time order parameter and β is the spatial fractal-order parameter. Here both x and X are vectors of dimension *N* and $x^{\beta }=\left(\right|x_{1}{|}^{\beta },|x_{2}{|}^{\beta },\ldots ,|x_{N}{|}^{\beta })$ (similar for X). In fact, Equation (Equation 15) solves the following *N* dimensional fractal time diffusion
(16)$$\frac{\partial u}{\partial t^{\alpha }}=D\nabla_{\beta }^{2}u.$$

Lastly, to derive the fractal Poisson equation resulting from Maxwell’s equations in Section 3, we consider a slightly different distance metric. The integer Poisson equation $\nabla^{2}\phi =-4\pi \rho$ holds when $\phi \left(x\right)=\int \frac{\rho (y)}{r}\;dy$, where r=||x−y||. In the fractal domain, this radial metric needs to follow a power law, and thus, for s≥0, we let
$$f=\frac{1}{r_{s}}\;\;\mathrm{where}\;r_{s}=\sqrt{|x_{1}{|}^{s}+|x_{2}{|}^{s}+{\left|x_{3}\right|}^{s}}.$$The fractal gradient of *f* is
(17)$$\nabla_{\beta }f=-\frac{s}{2\beta r_{s}^{3}}r^{s-\beta },$$
where $r^{s-\beta }=|x_{1}{|}^{s-\beta }i+|x_{2}{|}^{s-\beta }j+{\left|x_{3}\right|}^{s-\beta }k$.

The fractal Laplacian of *f* is
(18)$$\begin{array}{c} \nabla_{\beta }^{2}f=-\frac{s}{2\beta^{2}}\left(\frac{\left(s-\beta \right)\left(|x_{1}{|}^{s-2\beta }+|x_{2}{|}^{s-2\beta }+{\left|x_{3}\right|}^{s-2\beta }\right)}{{r_{s}}^{3}}-\frac{3s\left(|x_{1}{|}^{2\left(s-\beta \right)}+|x_{2}{|}^{2\left(s-\beta \right)}+{\left|x_{3}\right|}^{2\left(s-\beta \right)}\right)}{2{r_{s}}^{5}}\right) \end{array}$$

Please note that if $\beta =\frac{s}{2}$, then $\nabla_{\beta }^{2}f=0$ except at the origin, and this property is essential to establishing the fractal Poisson equation.

## 3. Maxwell’s Equations

The classical electromagnetic theory deals with electric and magnetic fields and their interactions through four fundamental partial differential equations known as Maxwell’s equations [46]. It is important to note that Maxwell’s equations do not uniquely determine electromagnetic fields, and other constitutive relations are needed to define the interaction within a material [47]. In this section, we present the fractal form of Maxwell’s equations. We then show how these equations can be inferred from maximizing entropy under certain fractional moment constraints in subsequent sections.

Consider the following modifications of Maxwell’s equations using fractal operators
(19)$$\nabla_{\beta }\cdot E=\frac{\rho }{\epsilon_{0}}$$
(20)$$\nabla_{\beta }\times E=-\frac{\partial B}{\partial t^{\alpha }}$$
(21)$$\nabla_{\beta }\cdot B=0$$
(22)$$\frac{1}{\mu_{0}}\left(\nabla_{\beta }\times B\right)=J+\epsilon_{0}\frac{\partial E}{\partial t^{\alpha }}$$
where E is the electric field, B is the magnetic flux, J is the current density, ρ is the charge density, $\epsilon_{0}$ is the vacuum permittivity, and $\mu_{0}$ is the vacuum permeability.

Here, Equation (Equation 19) represents Gauss’s law for electric fields, Equation (Equation 20) represents Faraday’s law of electromagnetic induction, Equation (Equation 21) represents Gauss’s law of magnetism, and Equation (Equation 22) represents the differential form of Ampere’s law. We obtain this fractal form of Maxwell’s equations, in terms of a scalar electrostatic potential ϕ and a magnetic vector potential A by introducing
(23)$$\begin{array}{cc} E & =-\nabla_{\beta }\phi -\frac{\partial A}{\partial t^{\alpha }}\\ \; & \\ B & =\nabla_{\beta }\times A. \end{array}$$

Then Equation (Equation 20) is satisfied since
(24)$$\begin{array}{cc} \nabla_{\beta }\times E & =\nabla_{\beta }\times \left(-\nabla_{\beta }\phi -\frac{\partial A}{\partial t^{\alpha }}\right)=-\nabla_{\beta }\times \nabla_{\beta }\phi -\nabla_{\beta }\times \frac{\partial A}{\partial t^{\alpha }}=\frac{\partial }{\partial t^{\alpha }}\nabla_{\beta }\times A=-\frac{\partial B}{\partial t^{\alpha }}. \end{array}$$

The divergent free B field in Equation (Equation 21) holds because
(25)$$\begin{array}{c} \nabla_{\beta }\cdot B=\nabla_{\beta }\cdot \left(\nabla_{\beta }\times A\right)=0. \end{array}$$

By considering the relationship between space and time in Equation (Equation 15), and dependent upon the form of the probability densities, it becomes possible to use entropy dynamics (see Section 4) as a means to validate the remaining two equations in Maxwell’s theory. We first note that
(26)$$\nabla_{\beta }\cdot E=\nabla_{\beta }\cdot \left(-\nabla_{\beta }\phi -\frac{\partial A}{\partial t^{\alpha }}\right)=-\nabla_{\beta }^{2}\phi -\nabla_{\beta }\cdot \frac{\partial A}{\partial t^{\alpha }}.$$

Thus, to satisfy Equation (Equation 19) we must have
(27)$$-\epsilon_{0}\nabla_{\beta }^{2}\phi -\epsilon_{0}\nabla_{\beta }\cdot \frac{\partial A}{\partial t^{\alpha }}=\rho .$$

Let us assume the Coulomb gauge $A=A^{\prime }-\nabla_{\beta }\Xi$ and $\phi =\phi^{\prime }+\frac{\partial \Xi }{\partial t^{\alpha }}$ such that A is fractal divergence free, i.e., $\nabla_{\beta }\cdot A=0.$ Then
(28)$$\nabla_{\beta }^{2}\Xi =\nabla_{\beta }\cdot \nabla_{\beta }\Xi =\nabla_{\beta }\cdot \left(A^{\prime }-A\right)=\nabla_{\beta }\cdot A^{\prime }-\nabla_{\beta }\cdot A=\nabla_{\beta }\cdot A^{\prime }.$$

Now for the case when $\nabla_{\beta }\cdot A=0,$ Equation (Equation 27) reduces to
(29)$$\nabla_{\beta }^{2}\phi =-\frac{\rho }{\epsilon_{0}}.$$

This fractal form of Gauss’s law in Equation (Equation 29) is obtained using the potential ϕ(x) given by
(30)$$\phi \left(x\right)=\frac{1}{4\pi \epsilon_{0}}\int \frac{\rho (x)}{\left||x-y|\right|}\;dy.$$

For η>0, let $r_{\eta }=\sqrt{|x_{1}{|}^{s}+|x_{2}{|}^{s}+{\left|x_{3}\right|}^{s}+\eta^{s}}$, $f_{\eta }=\frac{1}{r_{\eta }}$, and $\beta =\frac{s}{2}$. As in the integer case, the function ϕ(x) has the form
(31)$$\phi \left(x\right)=\int \frac{\rho (y)}{r_{\eta }}\;dy_{H},$$
where integration is conducted on the fractal domain. It can be shown that the fractal Laplacian of $f_{\eta }$ is
(32)$$\nabla_{\beta }^{2}\;f_{\eta }=-\frac{3\;\eta^{s}}{{\left(\eta^{s}+|x_{1}{|}^{s}+|x_{2}{|}^{s}+{\left|x_{3}\right|}^{s}\right)}^{5/2}}.$$

Using the fractal measure in Equation (Equation 5), we obtain
(33)$$\;\int \int \int \;\nabla_{\beta }^{2}f_{\eta }\;dV_{H}=A\;\int \int \int \;\nabla_{\beta }^{2}f_{\eta }\;x_{1}^{\alpha_{1}-1}x_{2}^{\alpha_{2}-1}x_{3}^{\alpha_{3}-1}dV,$$
where $A=\frac{8\;\pi^{\frac{\alpha_{1}}{2}+\frac{\alpha_{2}}{2}+\frac{\alpha_{3}}{2}}}{\Gamma \left(\frac{\alpha_{1}}{2}\right)\;\Gamma \left(\frac{\alpha_{2}}{2}\right)\;\Gamma \left(\frac{\alpha_{3}}{2}\right)}$. Since $\beta =\frac{s}{2}$ and η→0, from Equation (Equation 18), $\nabla_{\beta }^{2}f_{\eta }=0$, except at the origin. Let ψ be a test function and $S^{2}$ be the unit sphere, then
(34)$$A\;\int \int \int \;\nabla_{\beta }^{2}f_{\eta }\;x_{1}^{\alpha_{1}-1}x_{2}^{\alpha_{2}-1}x_{3}^{\alpha_{3}-1}dV=-\lim_{\eta \to 0}\underset{S^{2}}{\;\int \int \int \;}\frac{3\;A\;\eta^{s}\;x_{1}^{\alpha_{1}-1}\;x_{2}^{\alpha_{2}-1}\;x_{3}^{\alpha_{3}-1}}{{\left(\eta^{s}+|x_{1}{|}^{s}+|x_{2}{|}^{s}+{\left|x_{3}\right|}^{s}\right)}^{5/2}}\;\;\psi \left(x\right)\;dV$$

One can then establish the Poisson equation by expanding ψ(x) around the origin and integrating. Integrating the above equation is difficult, and an explicit form in terms of elementary functions may not exist. Thus, one may have to resort to numerical integration to obtain an exact solution. Figure 2 is a plot of the numerically obtained integral in Equation (Equation 34) demonstrating its convergence as η→0. When s=2,
β=1 and $\alpha_{1}=\alpha_{2}=\alpha_{3}=1$, the integer Poisson equation is retrieved and Equation (Equation 34) yields −4π as expected.

Using the relationships in Equation (Equation 23) and letting $\mu_{0}\epsilon_{0}=c^{-2}$ (*c* is the speed of light), we obtain the last of Maxwell’s equations, previously given by Equation (Equation 22), in the form
(35)$$\begin{array}{c} -\nabla_{\beta }^{2}A+\frac{1}{c^{2}}\frac{\partial }{\partial t^{\alpha }}\nabla_{\beta }\phi +\frac{1}{c^{2}}\frac{\partial }{\partial t^{\alpha }}\frac{\partial A}{\partial t^{\alpha }}=\mu_{0}J. \end{array}$$

We now split J into fractal divergence and curl-free components. Let $J=J_{T}+J_{L}$, where $J_{T}$ is transversal and $J_{L}$ solenoidal parts of J such that $\nabla_{\beta }\cdot J_{T}=0$ and $\nabla_{\beta }\times J_{L}=0.$ Assuming the Coulomb gauge condition, Equation (Equation 35) can be split into two equations
(36)$$\begin{array}{c} \frac{1}{c^{2}}\frac{\partial }{\partial t^{\alpha }}\left(\frac{\partial A}{\partial t^{\alpha }}\right)=\nabla_{\beta }^{2}A+\mu_{0}J_{T} \end{array}$$


(37)
$$\begin{array}{c} \mu_{0}J_{L}=\frac{1}{c^{2}}\frac{\partial }{\partial t^{\alpha }}\nabla_{\beta }\phi . \end{array}$$


Since Equation (Equation 36) consists of a fractal wave equation, we present a solution to this fractal wave equation to facilitate comparisons to the maximum entropy method given in Section 4.

We then let
(38)$$q\left(x,t\right)=Ce^{i\left(k\cdot x-wt\right)}$$
where $x=(x_{1},x_{2},x_{3})$, w=|k|v and *C* is a normalization constant. It is well known that *q* is a solution of the integer wave equation
$$\frac{1}{v^{2}}\frac{\partial^{2}q}{\partial t^{2}}=\nabla^{2}q.$$To extend the solution of the integer wave equation to obtain the solution of the fractal wave equation, consider
(39)$$u\left(x,t\right)=Ce^{i\left(k\cdot x^{\beta }-wt^{\alpha }\right)}$$
where, as before, $x=(x_{1},x_{2},x_{3})$, w=|k|v, and for β>0,
$x^{\beta }$ is defined component-wise as $x^{\beta }=\left(\right|x_{1}{|}^{\beta },|x_{2}{|}^{\beta },|x_{3}{|}^{\beta }).$ Then it follows that
(40)$$\begin{array}{c} \frac{\partial }{\partial t^{\alpha }}\frac{\partial u}{\partial t^{\alpha }}=-{C|k|}^{2}v^{2}e^{i\left(k\cdot x^{\beta }-wt^{\alpha }\right)} \end{array}$$
and, for j=1,2,3
(41)$$\begin{array}{c} \frac{\partial }{\partial x_{j}^{\beta }}\frac{\partial u}{\partial x_{j}^{\beta }}-Ck_{j}^{2}e^{i\left(k\cdot x^{\beta }-wt^{\alpha }\right)}. \end{array}$$

Thus, the fractal Laplacian of *u* is
(42)$$\begin{array}{cc} \nabla_{\beta }^{2}u=\sum_{j=1}^{3}\frac{\partial }{\partial x_{j}^{\beta }}\frac{\partial u}{\partial x_{j}^{\beta }} & =-\sum_{j=1}^{3}Ck_{j}^{2}e^{i\left(k\cdot x^{\beta }-wt^{\alpha }\right)}=-Ce^{i\left(k\cdot x^{\beta }-wt^{\alpha }\right)}\sum_{j=1}^{3}k_{j}^{2}=-C{\left|k\right|}^{2}e^{i\left(k\cdot x^{\beta }-wt^{\alpha }\right)}. \end{array}$$

Comparing Equations (Equation 40) and (Equation 42), we arrive at the fractal wave equation
(43)$$\frac{1}{v^{2}}\frac{\partial }{\partial t^{\alpha }}\frac{\partial u}{\partial t^{\alpha }}=\nabla_{\beta }^{2}u.$$

In general, for one-dimensional scalar functions f(x) and g(t) such that
(44)$$u\left(x,t\right)=Ce^{i\left(kf-\omega g\right)}$$
we obtain the one-dimensional fractal integer wave equation with source *S*
(45)$$\frac{\partial^{2}u}{\partial t^{2}}=D\frac{\partial^{2}u}{\partial x^{2}}+S$$
where the source function is
(46)$$\begin{array}{c} S=Ckve^{z}\left(v\left(\frac{d^{2}}{dx^{2}}f+k{\left(\frac{df}{dx}\right)}^{2}\right)-\left(\frac{d^{2}g}{dt^{2}}+kv{\left(\frac{dg}{dt}\right)}^{2}\right)\right) \end{array}$$
and z=kf(x)−ωg(t). Note, if f(x)=x, g(t)=t then S=0.

Now for the fractal wave equation given in Equation (Equation 43), the source *S* is
(47)$$S=Ckve^{z}\left(\frac{v}{\beta^{2}x^{2\beta -1}}\left(\left(1-\beta \right)\frac{\partial f}{\partial x}+x\frac{\partial^{2}f}{\partial x^{2}}+kx{\left(\frac{\partial f}{\partial x}\right)}^{2}\right)-\frac{1}{\alpha^{2}t^{2\alpha -1}}\left(\left(1-\alpha \right)\frac{\partial g}{\partial t}+t\frac{\partial^{2}g}{\partial t^{2}}+ktv{\left(\frac{\partial g}{\partial t}\right)}^{2}\right)\right)$$
and when $f\left(x\right)=x^{\beta }$, $g\left(t\right)=t^{\alpha }$, ω=kv and $D=v^{2}$ we have S=0. This relation is generalized in Section 4.1 for three-dimensional problems.

In the next section, we will arrive at the same wave equations using a completely different approach based on entropy maximization. The goal is to derive a probability density function that can be applied to stochastic homogenization of the electrostatic scalar potential (ϕ) and magnetic vector potential (A) from Maxwell’s equations for non-Gaussian stochastic processes.

## 4. Entropy Dynamics Approach

Modeling real-world problems is inherently challenging, with one of the most common obstacles being the presence of incomplete data sets or limited knowledge. Furthermore, experimental data frequently include noise, posing difficulties in obtaining precise measurements, especially when dealing with multiscale phenomena. These factors drive the investigation of the maximum entropy method, which aims to answer the question: given a partial state of knowledge, how can we construct the least biased model? E.T. Jaynes, a pioneering figure in the maximum entropy approach, dedicated much of his research to developing statistical inference methods and establishing constructive criteria for deriving probability distributions based on partial knowledge [48]. This method enables one to derive the most unbiased estimate possible from a given data set by constructing the most general model available. By identifying the most unbiased posterior densities, we can apply them to the electromagnetic theory of the previous section to construct models from available data and appropriate constraints.

The maximum entropy approach typically begins by defining a cost function that incorporates entropy and constraints derived from available data or observations. Normally, homogenization is conducted over all the possible states the systems may exhibit at any time instant. Here, we include the time dimension in the homogenization. This allows one to derive posteriors that may include conservative dynamics, i.e., wave equations, in addition to diffusive characteristics that do not conserve energy. This results in an entropy relation that is similar to action potentials that are used in Lagrangian electromagnetics and mechanics formulations [37]. The differences in conserved versus non-conserved energy relations will depend on the forms of the constraints that are reflected in the Lagrange multipliers and experimental observations.

We start with the following expression for entropy
(48)$$S=-\int p\left(x|X\right)\ln\left(\frac{p(x|X)}{q(x|X)}\right)\;dx\;dt$$
where p(x|X) is a time-dependent conditional probability density of a particle originally located at X that has moved to x=x(X,t) at time *t*. A prior density is also included, q(x|X), which is assumed to be flat or uninformative. We maximize the entropy *S* with the following constraints
(49)$$\begin{array}{cc} \; & \int p\left(x|X\right)\;dx\;dt=1\;\mathrm{and}\;\int p\left(x|X\right)f_{I}\left(x,t\right)\;dx\;dt=\sigma_{I} \end{array}$$
where the first constraint ensures unity of the probability for all time *t* and $f_{I}\left(x,t\right)$ includes three additional constraints that are associated with the respective moments $\sigma_{I}$ for I=1,2,3 that will be described later in this section. These moments must be determined from experimental observations or higher fidelity simulations.

Using the Lagrange multiplier method, the associated cost function is
(50)$$H=S-\kappa \left(\int p(x|X)\;dx\;dt-1\right)-\lambda_{I}\left(\int p\left(x|X\right)\;f_{I}\left(x,t\right)\;dx\;dt-\sigma_{I}\right).$$

We find the extremum of *H* with respect to the probability density p(x|X). To do so, we take functional derivatives of *H* with respect to p(x|X) and set it to zero according to
(51)$$\frac{\delta H}{\delta p}=\ln\left(\frac{p}{q}\right)+1+\kappa +\sum_{I=1}^{3}\left(\lambda_{I}f_{I}\left(x,t\right)\right)=0.$$

Simplifying the above equation gives us
(52)$$p\left(x|X\right)=q\left(x|X\right)e^{-1-\kappa -\sum_{I=1}^{3}\left(\lambda_{I}f_{I}\left(x,t\right)\right)}.$$

Since
(53)$$\begin{array}{c} \int p\;dx\;dt=\int qe^{-1-\kappa -\sum_{I=1}^{3}\left(\lambda_{I}f_{I}\left(x,t\right)\right)}\;dx\;dt=qe^{-1-\kappa }\int e^{-\sum_{I=1}^{3}\left(\lambda_{I}f_{I}\left(x,t\right)\right)}\;dx\;dt \end{array}$$
we let $Z=\frac{e^{1+\kappa }}{q}$ be partition function. Then
(54)$$p\left(x|X\right)=\frac{1}{Z}e^{-\sum_{I=1}^{3}\left(\lambda_{I}f_{I}\left(x,t\right)\right)}.$$

This probability density function is the result of incorporating constraints in the form of functions $f_{I}$. The crucial task lies in determining the appropriate constraints, which should be based on observations or relevant properties of the system and the quantities of interest. We propose two spatially dependent constraints and one time-dependent constraint that are used for comparisons to observable electromagnetic experimental data or high-fidelity simulations.

The three constraints include $f_{1}=k\cdot x^{\beta }$, $f_{2}=\omega t^{\alpha }$, and $f_{3}=c_{3}\ln\left(∥x∥\right)$ with α,β>0. Then
(55)$$p=\frac{1}{Z}e^{-\lambda_{1}k\cdot x^{\beta }-\lambda_{2}\omega t^{\alpha }-\lambda_{3}c_{3}\ln\left(∥x∥\right)}=\frac{1}{Z{∥x∥}^{\lambda_{3}c_{3}}}e^{-\lambda_{1}k\cdot x^{\beta }-\lambda_{2}\omega t^{\alpha }}.$$

To integrate the posterior *p*, we use spherical coordinates and assume spherically symmetric behavior. Specifically, we express the vector x as
x=(rsinθcosϕ,rsinθsinϕ,rcosθ).This gives ∥x∥=r, and the volume element becomes $dx=r^{2}\sin\theta \;dr\;d\theta \;d\phi$. Assuming spherical symmetry, we can write $k\cdot x^{\beta }=∥k∥r^{\beta }$. Then
(56)$$\begin{array}{cc} \int_{0}^{\infty }\int_{R^{3}}p\;dx\;dt & =\int_{0}^{\infty }\int_{R^{3}}\frac{1}{Z{∥x∥}^{\lambda_{3}c_{3}}}e^{-\lambda_{1}k\cdot x^{\beta }-\lambda_{2}\omega t^{\alpha }}dx\;dt\\ \; & =\int_{0}^{\infty }\int_{0}^{2\pi }\int_{0}^{\pi }\int_{0}^{\infty }\frac{1}{Zr^{\lambda_{3}c_{3}}}e^{-\lambda_{1}∥k∥r^{\beta }-\lambda_{2}\omega t^{\alpha }}r^{2}\sin\theta \;dr\;d\theta \;d\phi \;dt\\ \; & =4\pi \int_{0}^{\infty }\int_{0}^{\infty }\frac{1}{Zr^{\lambda_{3}c_{3}}}e^{-\lambda_{1}∥k∥r^{\beta }-\lambda_{2}\omega t^{\alpha }}r^{2}\;dr\;dt\\ \; & =\frac{4\pi }{Z}\frac{{\left(\lambda_{1}∥k∥\right)}^{\frac{\lambda_{3}c_{3}-3}{\beta }}{\left(\lambda_{2}\omega \right)}^{-1/\alpha }}{\alpha \beta }\Gamma \left(\frac{1}{\alpha }\right)\Gamma \left(\frac{3-\lambda_{3}c_{3}}{\beta }\right) \end{array}$$

Note we used $\int_{0}^{\infty }e^{-at^{b}}\;dt=\frac{a^{-\frac{1}{b}}}{b}\Gamma \left(\frac{1}{b}\right).$ Since ∫pdxdt=1 we get
(57)$$Z=\frac{4\pi {\left(\lambda_{1}∥k∥\right)}^{\frac{\lambda_{3}c_{3}-3}{\beta }}{\left(\lambda_{2}\omega \right)}^{-1/\alpha }}{\alpha \beta }\Gamma \left(\frac{1}{\alpha }\right)\Gamma \left(\frac{3-\lambda_{3}c_{3}}{\beta }\right)$$

Using the constraint $\int p\left(x|X\right)f_{2}\;dx\;dt=\omega \int p\left(x|X\right)t^{\alpha }\;dx\;dt=\sigma_{2}$ and the fact that
(58)$$\int_{0}^{\infty }t^{\gamma }e^{-\lambda_{2}\omega t^{\alpha }}\;dt=\frac{{\left(\lambda_{2}\cdot \omega \right)}^{-\frac{\gamma +1}{\alpha }}}{\alpha }\Gamma \left(\frac{\gamma +1}{\alpha }\right)$$
we get
(59)$$\begin{array}{c} \sigma_{2}=\int p\left(x|X\right)\omega t^{\alpha }\;dx\;dt=\frac{1}{Z}\left(\int_{R^{3}}\frac{1}{r^{\lambda_{3}c_{3}}}e^{-\lambda_{1}∥k∥r^{\beta }}r^{2}\;dr\right)\left(\int_{0}^{\infty }\omega t^{\alpha }\exp\left(-\lambda_{2}\omega t^{\alpha }\right)dt\right)=\frac{1}{\alpha \lambda_{2}}. \end{array}$$

Thus
(60)$$\lambda_{2}=\frac{1}{\alpha \sigma_{2}}.$$

Similarly, from constraint $\int p\left(x|X\right)f_{1}\;dx\;dt=\int p\left(x|X\right)\left(k\cdot x^{\beta }\right)\;dx\;dt=\sigma_{1}$, we get
(61)$$\begin{array}{cc} \int_{0}^{\infty }\int_{R^{3}}p\left(x|X\right)\left(k\cdot x^{\beta }\right)\;dx\;dt & =\frac{1}{Z}\int_{0}^{\infty }\int_{R^{3}}\frac{1}{{∥x∥}^{\lambda_{3}c_{3}}}e^{-\lambda_{1}k\cdot x^{\beta }-\lambda_{2}\omega t^{\alpha }}\left(k\cdot x^{\beta }\right)\;dx\;dt\\ \; & =\frac{4\pi }{Z}\int_{0}^{\infty }\int_{0}^{\infty }\frac{1}{r^{\lambda_{3}c_{3}}}\left(k\cdot x^{\beta }\right)\exp\left(-\lambda_{1}∥k∥r^{\beta }-\lambda_{2}\omega t^{\alpha }\right)r^{2}\;dr\;dt\\ \; & =\frac{4\pi }{Z}\left(\int_{0}^{\infty }\frac{k\cdot x^{\beta }}{r^{\lambda_{3}c_{3}}}\exp\left(-\lambda_{1}∥k∥r^{\beta }\right)r^{2}\;dr\right)\left(\int_{0}^{\infty }\exp\left(-\lambda_{2}\omega t^{\alpha }\right)dt\right)\\ \; & =\frac{1}{\lambda_{1}}\left(\frac{3-\lambda_{3}c_{3}}{\beta }\right). \end{array}$$

Thus
(62)$$\lambda_{1}=\frac{z}{\sigma_{1}}.$$
where $z=\frac{3-\lambda_{3}c_{3}}{\beta }$.

Using the third constraint $f_{3}=c_{3}\ln\left(∥x∥\right)$, we apply
(63)$$\int_{0}^{\infty }\int_{R^{3}}p\left(x|X\right)f_{3}\left(x\right)\;dxdt=\sigma_{3}.$$The last Lagrange multiplier $\lambda_{3}$ can be solved based upon
(64)$$\int_{0}^{\infty }\ln\left(r\right)r^{2-\lambda_{3}c_{3}}e^{-\lambda_{1}∥k∥r^{\beta }}dr=\frac{{\left(\lambda_{x}\;∥k∥\right)}^{-z}}{\beta }\left(\psi \left(z\right)-\ln\left(∥k∥\lambda_{1}\right)\right)$$
where ψ(z) is the Digamma function, and $z=\frac{3-\lambda_{3}c_{3}}{\beta }.$

The final posterior p(x|X) is
(65)$$p\left(x|X\right)=\frac{1}{Z{∥x∥}^{\lambda_{3}c_{3}}}\exp\left(-\frac{z}{\sigma_{1}}\left(k\cdot x^{\beta }\right)-\frac{\omega }{\alpha \sigma_{2}}t^{\alpha }\right).$$

We note that when the singularity constraint ln(∥x∥) is removed (i.e., when $\lambda_{3}=0$), the function *p* satisfies the 3D fractal wave equation given in Equation (Equation 43), where
(66)$$D={\left(\frac{\omega \beta \sigma_{1}}{3∥k∥\alpha \sigma_{2}}\right)}^{2}.$$

A plot of the one-dimensional version of the final posterior *p*, without the singularity constraint, which solves the fractal diffusion equation in 1D, is shown in Figure 3. In the plot, the fractal derivative order parameters α and β were chosen to illustrate how the probability density changes when varying the spatial moment $\sigma_{1}$, which can be extracted from available data. As expected, the probability density for data with higher moments tends to be much broader than for data that is more concentrated.

The relationship on the time-dependent Bayesian posterior can be more broadly applied to any scalar, vector, or tensor field using stochastic homogenization as previously done in Equation (Equation 1). In the following subsection, we apply Equation (Equation 65) to homogenize the scalar electrostatic ϕ and magnetic vector potentials A and relate the maximum entropy constraints to physical relations in the fractal form of Maxwell’s equations.

### 4.1. Homogenized Electromagnetic Potentials

Here, we illustrate how homogenized electromagnetic fields provide a method to construct a fractal version of Maxwell’s equations using various forms of the posterior given by Equation (Equation 65). The fractional-order constraints introduced in the previous section guide the fractal-order space–time operators and source terms based on available data. Upon presenting the self-consistent relationships, we highlight how maximum entropy methods may be used in practice to identify different forms of electromagnetic equations given sparse amounts of data.

We start with the homogenization of the electrostatic potential ϕ and magnetic vector potential A over the posterior density according to
(67)$$\phi \left(x,t\right)=\int_{\Omega_{0}}p\left(x|X\right)\phi \left(X,t_{0}\right)\;dX.$$
and
(68)$$A\left(x,t\right)=\int_{\Omega_{0}}p\left(x|X\right)A\left(X,t_{0}\right)\;dX.$$

The simplest case considers wave dynamics of A in Equation (Equation 68) for the limiting situation where the natural logarithm constraint ($f_{3}$) is neglected. In this case, $\lambda_{3}=c_{3}=0$ such that there are no singularities in the posterior given by Equation (Equation 65). In this case, it is easily shown that the fractal wave equation is self-consistent with the homogenized magnetic vector potential in Equation (Equation 68) and given by
(69)$$\begin{array}{c} \frac{\partial }{\partial t^{\alpha }}\left(\frac{\partial A}{\partial t^{\alpha }}\right)=D\nabla_{\beta }^{2}A \end{array}$$
which is equivalent to the free space form of Equation (Equation 36) when $D={\left(\frac{\omega \beta \sigma_{1}}{3∥k∥\alpha \sigma_{2}}\right)}^{2}=c^{2}$ and $J_{T}=0$.

Extending the wave equation for A to include the source term based on a non-zero transverse current density ($J_{T}\neq 0$) may occur dependent upon errors in matching the fractal derivative order exactly with the posterior density or the additional presence of the singularity in Equation (Equation 65). Below, we show the form of a source term that would be an additional term in Equation (Equation 69) when the fractal operator orders, α and β, do not match the power-law relations in the posterior density. Adding the singularity in the maximum entropy formulation, i.e., $\lambda_{3}\neq 0$ and $c_{3}\neq 0$, results in a highly complex source and is left for future numerical analysis. Generally speaking, the procedure to identify the non-zero source requires applying the same fractal time and space derivatives to Equation (Equation 68) and identifying the additional source term(s) that balance the wave equation. To illustrate the relationship between the wave equation for A and the homogenization from the posterior density, the source for the non-singular constraint is given as follows.

For a probability distribution that does not contain the singularity, the posterior has the form $p=Ce^{-f(x)-\omega g(t)}$, which satisfies the wave equation with *D* from Equation (Equation 66) along with the following source. Again, this form is based on the assumption that the fractal-order operators do not exactly match the power-law characteristics in the posterior. We also note that this scalar term given as follows is contained within the integral Equation (Equation 68) and would be multiplied by the magnetic vector A and integrated to explicitly relate it to $J_{T}$. This requires numerical analysis, as discussed previously, concerning numerically integrating Equation (Equation 34). In such cases, the scalar source term has the form
(70)$$S\left(x,t\right)=-\left(\frac{\omega \;t^{1-2\;\alpha }}{Z\;\alpha^{2}}\left(\left(\alpha -1\right)\frac{\partial g}{\partial t}-t\;\frac{\partial^{2}g}{\partial t^{2}}+\omega \;t\;{\left(\frac{\partial g}{\partial t}\right)}^{2}\right)+\frac{\omega^{2}\;{\sigma_{1}}^{2}\left(L_{1}+\left(1-\beta \right)L_{2}-L_{3}\right)}{9Z\;∥k∥\;\;\alpha^{2}\;{\sigma_{2}}^{2}\;{x_{1}}^{2\;\beta }\;{x_{2}}^{2\;\beta }\;{x_{3}}^{2\;\beta }}\right){Ce}^{-\;f-\omega \;g}$$
where
(71)$$L_{1}=\sum_{k=1}^{3}\left(x_{k}^{2}\;\frac{\partial^{2}f}{\partial {x_{k}}^{2}}\prod_{j\neq k}x_{j}^{2\;\beta }\right),\;\;L_{2}=\sum_{k=1}^{3}\left(x_{k}\;\frac{\partial f}{\partial x_{k}}\prod_{j\neq k}x_{j}^{2\;\beta }\right),\;\mathrm{and}\;L_{3}=\sum_{k=1}^{3}\left(x_{k}^{2}\;{\left(\frac{\partial f}{\partial x_{k}}\right)}^{2}\prod_{j\neq k}x_{j}^{2\;\beta }\right).$$As expected, this source vanishes when the observations follow a power law with the form $f=k\cdot x^{\beta }$ and $g\left(t\right)=t^{\alpha }.$ The more general form containing the singularity is more complex and will most likely require additional numerical analysis.

We now derive the case for Gauss’s law Equation (Equation 29), which contains no explicit time-dependence. In this case, the second constraint is neglected where $\lambda_{2}=0$. Physically, this can be interpreted by $\sigma_{2}$, which constitutes a measure of the period of time-varying oscillations of charged particles. If $\sigma_{2}\to \infty$, no oscillations exist and $\lambda_{2}\to 0$ as seen from Equation (Equation 60). In this case, the posterior Equation (Equation 65) contains no time-dependence but does include the singularity. For clarity, the posterior for the electrostatic potential is then
(72)$$p_{\phi }\left(x|X\right)=\frac{1}{Z{∥x∥}^{\lambda_{3}c_{3}}}\exp\left(-\frac{z}{\sigma_{1}}\left(k\cdot x^{\beta }\right)\right).$$

We now apply $p_{\phi }\left(x|X\right)=p\left(x|X\right)$ to the electrostatic homogenization in Equation (Equation 67) and obtain an explicit relationship for the fractal Laplacian, $\nabla_{\beta }^{2}p_{\phi }=S\left(x\right)$. We determine the source S(x) to be associated with certain characteristics of the posterior. We first note that this source is associated with the charge density ρ(x) given in Equation (Equation 29). Now consider the simpler case where $\lambda_{3}=0$. In this case, a source term depends on the fractional variance given by $\sigma_{1}$. In the case where this fractional variance is very large, it is expected that ρ(x)→0. Physically, this means the stochastic homogenization of charges balances exactly and no internal electric fields are acting on the charges. This assessment is based on an ideal infinite medium and thus excludes the possibility of charges on a boundary. When internal charges are unequally distributed, the imbalance in charge leads to $\sigma_{2}\neq 0$, a non-zero charge density ρ(x), and non-zero internal fields. The source term based on the fractal Laplacian of $p_{\phi }$, excluding the singularity, is
(73)$$S\left(x\right)=\frac{9\;D\;{∥k∥}^{2}}{Z\;\beta^{2}\;{\sigma_{1}}^{2}}\;\exp\left(-\frac{3}{\beta \;\sigma_{1}}\left(k\cdot x^{\beta }\right)\right).$$

The singularities in the posterior where $\lambda_{3}$ and $c_{3}$ create additional effects governing the balance of charge density and complexities in the internal electric fields. These power-law singularities may describe a refined force field from charged interactions between particles, which vary as a function of the surrounding particle displacements. Given that this is a homogenized relationship based on the posterior, it gives an estimate of many-body interactions from all the surrounding charges. An assessment of its importance requires numerical analysis to compute the charge density from the source S(x).

### 4.2. Self-Consistent Fractal Charge Conservation and Field Relations

Given the complexities of quantifying non-Gaussian Bayesian posterior densities and relating them to the fractal electromagnetic equations, we give an example illustrating where fractal operators become important in quantifying electromagnetic interactions in complex materials. We start with the conservation of charge and illustrate when fractal operators are superior in predicting charge flux and electromagnetic fields through a fractal representative volume element (RVE). We then compare the fractal conservation of charge with the fractal form of Maxwell’s equations to illustrate self-consistency in the electromagnetic field equations.

Similar to the derivation of fractional conservation of mass in porous or fractal media [16], we start by assuming the flux of total charge $J=J_{T}+J_{L}$ varies nonlinearly through an RVE. Given that fractal geometry follows a power-law function, we assume the flux of charge follows a power-law relationship due to fractal geometric constraints. The analysis is different from the nonlocal fractional conservation of mass given by Wheatcraft and Meerschaert [16]. The fractal operator is more robust in approximating power-law functions due to the invariance of the operating point chosen in the fractal Taylor series (see p. 6 [19]). However, the basis functions in fractional or fractal Taylor expansions are different and, therefore may affect the accuracy of predictions of data. Lastly note that we also generalize the charge conservation to include fractal operators, not only in space but also in time.

The problem is illustrated in Figure 4 where a RVE of charge is assumed to have a nonlinear flux over the volume $\Delta V=\Delta x_{1}\Delta x_{2}\Delta x_{3}$. In such cases, we assume the current density follows a power-law function over space. For example, we denote the component of the current density ($J_{1}$) in the $x_{1}$ direction to vary in space according to
(74)$$J_{1}=\bar{J}+\tilde{J}\Delta x_{1}^{\nu }$$
where $\Delta x_{1}^{\nu }=|x_{1}+\Delta x_{1}{|}^{\nu }-{\left|x_{1}\right|}^{\nu }$. $\bar{J}$ and $\tilde{J}$ are phenomenological constants and ν is the exponent.

Using the first-order fractal Taylor expansion of order β to estimate Equation (Equation 74), we have
(75)$$J_{1}\left(x_{1}+\Delta x_{1},x_{2},x_{3}\right)=J_{1}\left(x_{1},x_{2},x_{3}\right)+{\left.\frac{\partial J_{1}}{\partial x_{1}^{\beta }}\right|}_{x_{1},x_{2},x_{3}}\Delta x_{1}^{\beta }$$
where it can be shown using the fractal operator definitions in Section 2.2 that this perfectly matches Equation (Equation 74) with one fractal derivative if β=ν.

We can generalize this relationship to 3D to obtain the summation of flux gradients in the $x_{1},x_{2}$ and $x_{3}$ directions. We first denote $\Delta J_{1}=J_{1}\left(x_{1}+\Delta x_{1},x_{2},x_{3}\right)-J_{1}\left(x_{1},x_{2},x_{3}\right)$ (and similar for $\Delta J_{2},\Delta J_{3}$) to write out the complete fractal Taylor expansion in 3D as
(76)$$\Delta x_{2}^{\beta }\Delta x_{3}^{\beta }\Delta J_{1}+\Delta x_{1}^{\beta }\Delta x_{3}^{\beta }\Delta J_{2}+\Delta x_{1}^{\beta }\Delta x_{2}^{\beta }\Delta J_{3}={\left.\frac{\partial J_{i}}{\partial x_{i}^{\beta }}\right|}_{x_{1},x_{2},x_{3}}\Delta V^{\beta }$$
where a summation is implied on the index *i*.

For the fractional conservation of mass [16], it was assumed that the changes in mass flux over an RVE balance with an internal change in mass are dependent upon the integer time derivative of mass density. Since it is assumed that charges vary nonlinearly in space due to their fractal characteristics of the underlying material structure, we also assume that their *time* rate of change also varies nonlinearly according to a power-law function. This relation is described in terms of the total charge
(77)$$Q\left(x,t\right)=\bar{Q}\left(x\right)+\tilde{Q}\left(x\right)\Delta t^{\gamma }$$
where $\Delta t^{\gamma }=t^{\gamma }-t_{0}^{\gamma }$ where $t_{0}$ is some initial time. $\bar{Q}$ and $\tilde{Q}$ are phenomenological parameters and γ is the exponent on time increments.

Since we focus on the Eulerian (constant volume) frame, we take the partial fractal time derivative of the total charge
(78)$$Q\left(x,t+\Delta t\right)=Q\left(x,t\right)+{\left.\frac{\partial Q(x,t)}{\partial t^{\alpha }}\right|}_{t}\Delta t^{\alpha }$$
of order α. If γ=α, the fractal Taylor expansion over time perfectly matches the relationship given by Equation (Equation 77).

We can now balance the fractal flux across the RVE in Equation (Equation 76) with the fractal time rate of change in total charge as
(79)$$\frac{\partial Q}{\partial t^{\alpha }}=-\frac{\partial J_{i}}{\partial x_{i}^{\beta }}\Delta V^{\beta }$$
and note the sign difference accounts for positive charge accumulation in the RVE is in the opposite direction of the outward pointing unit normal. This could also be re-written by normalizing the total charge *Q* with the fractal volume element $\Delta V^{\beta }$ to obtain the charge density over the fractal RVE to give
(80)$$\frac{\partial \rho }{\partial t^{\alpha }}+\frac{\partial J_{i}}{\partial x_{i}^{\beta }}=0$$
where $\rho =\lim_{\Delta V\to 0}\frac{Q}{\Delta V^{\beta }}$.

Given the fractal conservation of charge in Equation (Equation 80), we highlight its self-consistency with the fractal version of Maxwell’s equations. Starting with Ampere’s law in Equation (Equation 22), we take the fractal divergence to obtain
(81)$$\nabla_{\beta }\cdot \frac{1}{\mu_{0}}\left(\nabla_{\beta }\times B\right)=\nabla_{\beta }\cdot J+\epsilon_{0}\nabla_{\beta }\cdot \frac{\partial E}{\partial t^{\alpha }}$$
and due to the linearity of the fractal operators, $\nabla_{\beta }\cdot \frac{1}{\mu_{0}}\left(\nabla_{\beta }\times B\right)=0$, we have
(82)$$\nabla_{\beta }\cdot J+\epsilon_{0}\nabla_{\beta }\cdot \left(\frac{\partial E}{\partial t^{\alpha }}\right)=0.$$

Since the order of the fractal time and space operators is invariant, we can relate the fractal divergence of the electric field to the fractal Gauss’ law that was given by Equation (Equation 19). We then arrive at a self-consistent relationship with the fractal conservation of charge in Equation (Equation 80). This illustrates that if fractal geometry imposes constraints on the flux of charge in the form of power-law nonlinearities in both space and time, fractal operators are more accurate than integer operators in predicting changes in the charge distribution, which also influences the fractal form of the electromagnetic field equations.

### 4.3. Fractional Unit Scaling

Questions arise in interpreting the fractal-order electromagnetic field equations due to the non-intuitive fractional-order units that result from these space–time operators. Similar issues arise in nonlocal fractional-order operators. The interpretation of these operators and units has been discussed in the literature [24,49,50,51,52,53]. For example, Tarasov [24] normalizes material coordinates such that all units associated with length are unitless. Questions remain when applying such equations to experiments and inferring material parameters that include length. Gómez-Aguilar [49] introduced a pre-factor into the fractal operator that cancels the fractional unit in the fractal operator. They apply this operator to resistor-capacitor circuits. We consider their method here for the electromagnetic field equations as one alternative to avoid questions surrounding interpretations of fractional space and time units.

Following [49], the fractal operator is modified by the pre-factor using
(83)$$\begin{array}{c} \frac{\partial }{\partial t^{\alpha }}\to \frac{1}{\sigma_{t}^{1-\alpha }}\frac{\partial }{\partial t^{\alpha }} \end{array}$$
(84)$$\begin{array}{c} \frac{\partial }{\partial x_{i}^{\beta }}\to \frac{1}{\sigma_{x}^{1-\beta }}\frac{\partial }{\partial x_{i}^{\beta }} \end{array}$$
where $\sigma_{t}$ and $\sigma_{x}$ are parameters with units of time [s] and length [m], respectively.

We first apply this pre-factor to the electromagnetic field relations previously given by Equation (Equation 23) followed by Maxwell’s equations in terms of ϕ and A as previously given by Equations (Equation 29) and (Equation 36), respectively. We include this modification here to avoid overcomplicating the prior equations with two additional parameters.

We first illustrate the scaling on the electric field and magnetic flux density by applying
(85)$$\begin{array}{cc} E_{i} & =-\frac{\partial \phi }{\partial x_{i}^{\beta }}-\frac{\partial A_{i}}{\partial t^{\alpha }}\to -\frac{1}{\sigma_{x}^{1-\beta }}\frac{\partial \phi }{\partial x_{i}^{\beta }}-\frac{1}{\sigma_{t}^{1-\alpha }}\frac{\partial A_{i}}{\partial t^{\alpha }} \end{array}$$
(86)$$\begin{array}{cc} B_{i} & =e_{ijk}\frac{\partial A_{k}}{\partial x_{j}^{\beta }}\to \frac{1}{\sigma_{x}^{1-\beta }}e_{ijk}\frac{\partial A_{k}}{\partial x_{j}^{\beta }}. \end{array}$$
By inspection, if the units of the parameters $\sigma_{t}$ and $\sigma_{x}$ are time [s] and length [m], respectively, the electric field units are voltage/length [V/m] and the magnetic flux units are Tesla [T].

It can also be shown that the units remain consistent in Maxwell’s equations when written in terms of ϕ and A. For completeness, we show
(87)$$\begin{array}{cc} \frac{\partial^{2}\phi }{\partial x_{i}^{\beta }} & =-\frac{\rho }{\epsilon_{0}}\to \frac{1}{\sigma_{x}^{2(1-\beta )}}\frac{\partial^{2}\phi }{\partial x_{i}^{\beta }}=-\frac{\rho }{\epsilon_{0}} \end{array}$$
(88)$$\begin{array}{cc} \frac{1}{c^{2}}\frac{\partial }{\partial t^{\alpha }}\left(\frac{\partial A_{i}}{\partial t^{\alpha }}\right) & =\frac{\partial }{\partial x_{i}^{\beta }}\left(\frac{\partial A_{i}}{\partial x_{i}^{\beta }}\right)+\mu_{0}J_{T}\to \frac{1}{c^{2}\sigma_{t}^{2(1-\alpha )}}\frac{\partial }{\partial t^{\alpha }}\left(\frac{\partial A_{i}}{\partial t^{\alpha }}\right)=\frac{1}{\sigma_{x}^{2(1-\beta )}}\frac{\partial }{\partial x_{i}^{\beta }}\left(\frac{\partial A_{i}}{\partial x_{i}^{\beta }}\right)+\mu_{0}J_{T} \end{array}$$
which again illustrates how the introduction of the parameters $\sigma_{t}$ and $\sigma_{x}$ with units of [s] and [m] give integer units in each equation. It should be noted, however, that the values of these scaling parameters must be carefully inferred from experiments or high-fidelity simulations so that they do not violate physical limits such as the speed of light. We comment further on this and other invariance relationships between the Euclidean and fractal domains in the following section.

## 5. Discussion

An entropy dynamics framework has been developed and applied to Maxwell’s time-dependent electromagnetic equations. Fractional functions were introduced as constraints and balanced against a form of Shannon entropy that included integration over both space and time. The optimization of the cost led to a posterior that included effects associated with wave propagation of electromagnetic fields and divergent characteristics of electrostatics and magnetostatics. For example, we illustrate that when fractional constraints coincide with observations, the electromagnetic balance equations are self-consistent with fractal operators in both space and time. The additional sources in the fractal electromagnetic equations were shown to be associated with power-law singularities associated with multiscale interactions of charged particles as described in the posterior density.

The theoretical relations provide a method to relate fractal operators with non-Gaussian posterior densities given observations that match the proposed fractional moment constraints over space and time. These three constraints, first defined by Equation (Equation 49) and later by Equation (Equation 55), must be informed by measurements or higher fidelity simulations. The first constraint $\sigma_{1}$ provides a relative distance measure between material particles. From a molecular dynamics framework, these relative particle distances can be computed directly from atomistic simulations to quantify the statistical moment of a collection of atoms. The second constraint, $\sigma_{2}$, represents the time period of oscillations between material particles. Similar to equilibrium particle distances in the first constraint, the distribution of the period of oscillations can be computed from atomistic simulations or experimentally, such as through NMR measurements [54]. Lastly, the third constraint denoted by $\sigma_{3}$ is more complicated as it estimates longer-range spatial singularities from particle interactions. These effects give a refined force field between charged particles based upon the homogenization of long-range interactions over complex atomic distributions that may exhibit fractal characteristics. Current research is focused on conducting atomistic scale simulations for comparisons to the homogenized fields and fractal forms of Maxwell’s equations to build continuum surrogate models that accommodate complexities that are otherwise not well approximated using Gaussian densities.

The information-theoretic methods described here can inform materials scientists and design engineers about complex material systems and structures. For example, temporal and spatial data may result in non-Gaussian statistical distributions, making it challenging to simulate and predict nonlinear behavior with integer-order operators. Fractal-order operators can improve homogenized estimates of extreme events associated with the onset of nonlinearities originating at defects which become important in understanding material and device reliability. This provides a homogenized estimate of extreme events in materials, which can be useful in complementing large-scale molecular dynamic simulations that exhibit uncertainty in their repeatability due to chaotic particle dynamics [55]. The statistical quantification of particle interactions in terms of moments provides a direct feedback loop to the fractal electromagnetic equations to help understand deviations from linearity in the electromagnetic fields based on the underlying fractal material structure. This also has implications for device design in nanotechnology. For example, physical reservoir computers (PRC) have gained attention due to their novel ability to process information for artificial intelligence applications in controlling robotic and aerospace systems and structures [56,57,58]. While promising, high-performance PRCs require specific amounts of nonlinearity to process information efficiently. The proposed information-theoretic framework may facilitate the structure-property design problem to create ideal nonlinearities for embedded sensors within a physical reservoir structure to sense and assess the environment for more efficient feedback control.

An additional advantage of using the entropy dynamics framework is related to the constraints based on experimental observation, high-fidelity molecular dynamics, or quantum simulations. We highlighted in Section 4.2 how the conservation of charge and the associated electromagnetic fields can be more accurately estimated with fractal operators if power-law constitutive relationships exist. However, we have not addressed how transformations between the fractal domain and Euclidean domain affect invariance relationships (e.g., translational and rotational). Non-trivial relationships between Eulerian (deformed) and Lagrangian (undeformed) configurations have been rigorously evaluated to understand invariance conditions in the Euclidean domain, see [37], for example. To the author’s knowledge, this remains an open question on the fractal domain. However, given the constraints from observation within the entropy dynamics framework, limits on the fractal order should facilitate understanding the invariance relations. Nonetheless, transformations between the Euclidean and fractal domains should be further investigated to ensure that conservation properties are preserved.

## Figures and Tables

**Figure 1 entropy-26-01103-f001:**
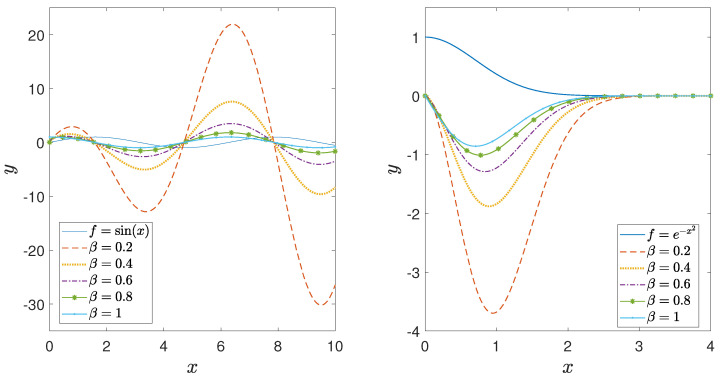
Fractal derivative of y=sin(x) and $y=e^{-x^{2}}$ for different values of β.

**Figure 2 entropy-26-01103-f002:**
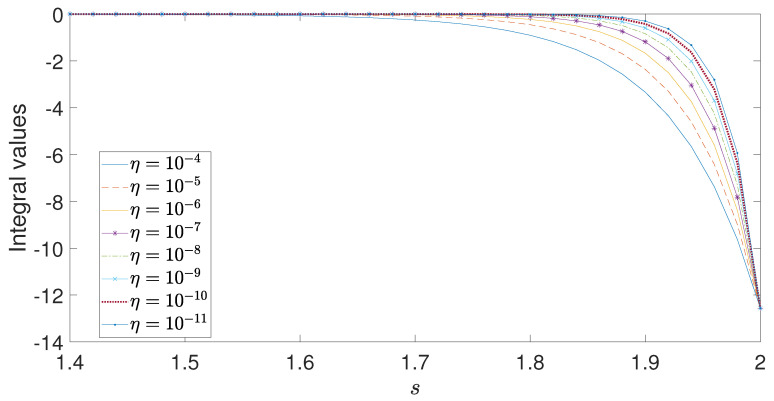
Plot of the integral in Equation (Equation 34) for decreasing η values and $\alpha_{1}=\alpha_{2}=\alpha_{3}=1$.

**Figure 3 entropy-26-01103-f003:**
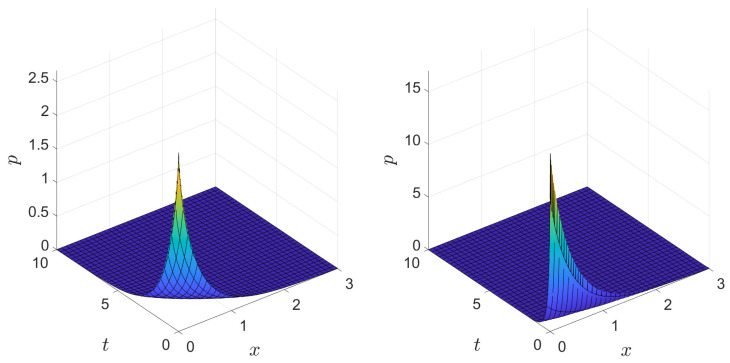
Plot of the posterior *p* for $\sigma_{1}=1.1,\sigma_{2}=0.5,\beta =0.9,$ and α=0.8 (**left**). Plot of the posterior *p* for $\sigma_{1}=0.3,\sigma_{2}=0.5,\beta =0.7$, and α=0.9 (**right**).

**Figure 4 entropy-26-01103-f004:**
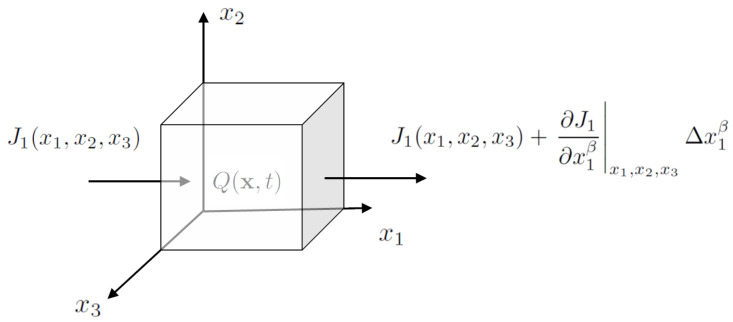
Illustration of flux and charge accumulation within a representative volume element (RVE) using fractal operators. The volume element has size $\Delta V=\Delta x_{1}\Delta x_{2}\Delta x_{3}$. The current density in the $x_{1}$ direction ($J_{1}$) over $\Delta x_{1}$ is shown along with Q(x,t) as the total charge in the RVE.

## Data Availability

Given the theoretical nature of this work no data is available. Interested researchers are encouraged to contact the corresponding author with questions.

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
