# Peer review of "An Entropy Dynamics Approach to Inferring Fractal-Order Complexity in the Electromagnetics of Solids"

_entropy, 2024, doi:10.3390/e26121103_

Round 1

Reviewer 1 Report

Comments and Suggestions for Authors

Clarity and Structure:

The paper introduces a novel fractal-order entropy dynamics model; however, some sections lack clarity. Please refine the presentation of the mathematical framework, particularly the derivations of the fractal derivative relationships with divergence, curl, and Laplacian operators, to make them more comprehensible to a broader audience.

Theoretical Justification:

While the paper integrates Shannon's entropy and Bayesian inference effectively, it would benefit from a more detailed theoretical justification for using fractal derivatives in the context of Maxwell's equations. Elaborating on the physical significance of fractal order and its relevance to the homogenization of electromagnetic fields would enhance the impact.

Numerical Validation and Examples:

The study lacks numerical simulations or illustrative examples to validate the proposed model. Including case studies or computational results demonstrating the applicability of the fractal Maxwell equations to real-world problems (e.g., electromagnetic fields in solids) would strengthen the paper.

Applications and Implications:

The discussion on practical implications is relatively brief. Expanding on potential applications in areas such as material science, nanotechnology, or electromagnetic field modeling in heterogeneous media would make the work more appealing to readers from diverse fields.

Literature Integration:

The paper could better position itself in the context of existing research. Adding a comparison with conventional approaches to homogenizing electromagnetic fields, along with citing relevant studies on fractal derivatives and entropy dynamics, would provide a stronger foundation for the proposed methodology. 

Modified introduction with the help of these papers "Approximate analytical methods for a fractional-order nonlinear system of Jaulent–Miodek equation with energy-dependent Schrödinger potential" "An efficient analytical approach for the solution of certain fractional-order dynamical systems" "Fractional series solution construction for nonlinear fractional reaction-diffusion Brusselator model utilizing Laplace residual power series" "Fractional view analysis of Kersten-Krasil’shchik coupled KdV-mKdV systems with non-singular kernel derivatives"

Formatting and Language:

While the manuscript is well-written overall, minor grammatical errors and formatting inconsistencies detract from readability. Proofreading and adhering to the journal's formatting guidelines will improve the manuscript's quality.

By addressing these points, the paper will achieve greater clarity, stronger validation, and broader relevance to the scientific community.

Comments on the Quality of English Language

Check grammatical mistakes

Reviewer 2 Report

Comments and Suggestions for Authors

The authors present a clever and novel use of fractional Maxwell equations combined with an entropy dynamics approach to model some electromagnetic behavior in solids. The paper is well structured, interesting, and enjoyable to read. I support publication but I think there are two related and important things to address. I also have two very minor points.

1) While the authors are clear this a model, I think something must be said about units. I wish I could be a bit more helpful but this is something I have some difficulty thinking about when I work with fractional derivatives. First of all, some notion what a fractional meter or fractional second means should be given even it is not completely satisfying. More importantly, fundamental inconsistencies must be cleared up. There are a few spots where this should be done but the best example is in the consideration of the Fourier components in Eq. (30). I think one must require the argument of the exponential to be unitless. This would imply that k should have the complementary fractional wavenumber unit: m^-\beta. Likewise, the \omega should have the complementary fractional frequency unit: s^-\alpha. If that is the case I think it renders the equation \omega = |k|v inconsistent unless v is a fractional velocity but I think this cascades its why through other physical relationships. 

2) Related to point 1. I think the fractional Maxwell equations are no longer relativistically invariant in the normal way. Again, I think one for fractionalize the Lorentz transformation but that, too, will have cascade effects. I don't think relativistic invariance is necessary for the desired applications of this model but this issue should probably be addressed. 

Very Minor Points

3) Most of the displayed equations are numbered but some are not. Some of those numbered are not referenced in the body of the text. I think there are two typical conventions, either number all the displayed equations or number only those referred to in the text. 

4) In the text, equations are referenced such as "..as shown in (XX)" Maybe should read "..as shown in Eq. (XX)" I'm sure the editors can make the call on this. 

Round 2

Reviewer 2 Report

Comments and Suggestions for Authors

The authors addressed my concerns well.